# RL2 Enhances the Elimination of Breast Cancer Cells by Doxorubicin

**DOI:** 10.3390/cells12242779

**Published:** 2023-12-06

**Authors:** Fabian Wohlfromm, Kamil Seyrek, Nikita Ivanisenko, Olga Troitskaya, Dagmar Kulms, Vladimir Richter, Olga Koval, Inna N. Lavrik

**Affiliations:** 1Translational Inflammation Research, Medical Faculty, Center of Dynamic Systems (CDS), Otto von Guericke University, 39120 Magdeburg, Germany; fabian.wohlfromm@med.ovgu.de (F.W.); kamil.seyrek@med.ovgu.de (K.S.); nikita.ivanisenko@med.ovgu.de (N.I.); or o.troitskaya@beatson.gla.ac.uk (O.T.); 2Experimental Dermatology, Department of Dermatology, TU-Dresden, 01307 Dresden, Germany; dagmar.kulms@ukdd.de; 3National Center for Tumor Diseases, TU-Dresden, 01307 Dresden, Germany; 4Department of Biotechnology, Institute of Chemical Biology and Fundamental Medicine, Siberian Branch of Russian Academy of Sciences (SB RAS), 630090 Novosibirsk, Russia; richter@niboch.nsc.ru (V.R.); o.koval@niboch.nsc.ru (O.K.)

**Keywords:** RL2, breast cancer, DXR, mitophagy, intrinsic cell death, lactaptin, Κ-Casein

## Abstract

RL2 (recombinant lactaptin 2), a recombinant analogon of the human milk protein Κ-Casein, induces mitophagy and cell death in breast carcinoma cells. Furthermore, RL2 was shown to enhance extrinsic apoptosis upon long-term treatment while inhibiting it upon short-term stimulation. However, the effects of RL2 on the action of chemotherapeutic drugs that induce the intrinsic apoptotic pathway have not been investigated to date. Here, we examined the effects of RL2 on the doxorubicin (DXR)-induced cell death in breast cancer cells with three different backgrounds. In particular, we used BT549 and MDA-MB-231 triple-negative breast cancer (TNBC) cells, T47D estrogen receptor alpha (ERα) positive cells, and SKBR3 human epidermal growth factor receptor 2 (HER2) positive cells. BT549, MDA-MB-231, and T47D cells showed a severe loss of cell viability upon RL2 treatment, accompanied by the induction of mitophagy. Furthermore, BT549, MDA-MB-231, and T47D cells could be sensitized towards DXR treatment with RL2, as evidenced by loss of cell viability. In contrast, SKBR3 cells showed almost no RL2-induced loss of cell viability when treated with RL2 alone, and RL2 did not sensitize SKBR3 cells towards DXR-mediated loss of cell viability. Bioinformatic analysis of gene expression showed an enrichment of genes controlling metabolism in SKBR3 cells compared to the other cell lines. This suggests that the metabolic status of the cells is important for their sensitivity to RL2. Taken together, we have shown that RL2 can enhance the intrinsic apoptotic pathway in TNBC and ERα-positive breast cancer cells, paving the way for the development of novel therapeutic strategies.

## 1. Introduction

Breast cancer is one of the most frequent human cancers [1,2,3]. One of the major features of breast cancer is its heterogeneous nature due to the dysregulation of several proteins and genes, as well as mutations in key proteins such as p53 [4,5,6]. Three different types of invasive breast cancers have been defined: ERα-positive, HER2-positive, and TNBCs [2]. All three cancers have metastatic potential; however, they were shown to possess different relapse times and death rates [7].

Until today, one of the most commonly used therapies is the removal of breast cancer by surgery, followed by chemotherapy and/or radiotherapy. However, the acquired mutations can lead to resistance to chemotherapeutic drugs [8]. Additionally, relapsed breast cancer cells tend to metastasize and become more resistant, leading to a poor prognosis [9]. Specific therapies are being developed to treat breast cancer. For example, HER2-positive breast cancers are treated by administration of antibodies against the receptor, such as trastuzumab [7]. However, the development of resistance to conventional therapies and metastasis remain unsolved problems in the treatment of breast cancer, highlighting the need for more specific therapeutic strategies.

Doxorubicin (DXR) is a genotoxic drug that has been used since 1969 as a therapeutic agent in several cancer types, including breast cancer [10]. Induction of cell death via the intrinsic apoptotic pathway is one of the major modes of DXR action. However, DXR treatment is associated with cardiotoxicity [11,12]. This results in a limitation of the dose concentration and leads to severe side effects upon the increase in DXR dose [13].

The human milk protein Κ-casein possesses proteolytic fragments, such as lactaptin [14]. Based on lactaptin, different analogs, including RL1 (recombinant lactaptin 1) or RL2 (recombinant lactaptin 2), have been generated and shown to induce cell death in breast carcinoma cells but not in the non-cancerous cells [14,15] with RL2, which consists of amino acids 23–134 of human Κ–casein, being the strongest inducer of cell death [14]. RL2 treatment can lead to cell death in different breast carcinoma cells but not in normal cells and inhibit tumor growth in mouse models [16,17]. RL2 was shown to act in monomeric, dimeric, and oligomeric forms, which are formed via cysteine bridges [18,19]. It is still unknown how the cytotoxic activity of RL2 is controlled by the different degrees of RL2 oligomerization.

Recently, RL2 has been shown to induce autophagy and, in particular, mitophagy [17,20,21]. Specifically, it was shown that RL2 treatment caused an upregulation of PINK1, BNIP3, and BNIP3L/NIX and the conversion of LC3 to LC3 II in breast cancer cells [21]. In addition, it was shown that RL2 induced the loss of mitochondrial potential and ATP loss in breast cancer cells [19]. It was also shown that RL2 had no effect on oxygen consumption, indicating that the effect of RL2 does not directly involve perturbation of the mitochondrial respiratory chain [21]. Mass spectrometry analysis of the RL2 interactome has revealed an interaction of RL2 with the mitochondrial import protein TOM70 [19]. The RL2/TOM70 interaction was suggested to play a major role in RL2-induced cell viability loss and mitophagy [19,21]. Moreover, RL2 treatment was shown to control TRAIL-induced apoptosis [21]. In this context, the opposing effects of RL2/TRAIL co-treatment can be detected: upon short-term stimulation, RL2 inhibited the TRAIL-mediated cell death via induction of mitophagy, which limited the amount of death-inducing signaling complex (DISC) formation [21]. Upon long-term co-treatment, RL2 led to a sensitization towards TRAIL-induced cell death [21].

In contrast, the influence of RL2 on the action of chemotherapeutic drugs that induce the intrinsic apoptosis pathway in breast cancer cells remains to be determined. The latter may present a promising direction aiming to reduce the dose of chemotherapeutic agents and thus protect patients from severe side effects. In the current study, we analyzed the RL2 effects on DXR-induced cell death to further explore the molecular mechanisms of RL2 action on intrinsic pathways and pave the way toward new therapeutic strategies.

## 2. Materials and Methods

### 2.1. Antibodies and Reagents

All chemicals were of analytical grade and purchased from AppliChem (Darmstadt, Germany), Carl Roth (Karlsruhe, Germany), Gibco™ (Carlsbad, CA, USA), Merck (Darmstadt, Germany), Sigma-Aldrich (Taufkirchen, Germany), Santa Cruz Biotechnology (Heidelberg, Germany) or Thermo Fisher Scientific Inc. (Waltham, MA, USA). RL2 was purified as described previously [17]. Doxorubicin (DXR) (#10512955) was purchased from Thermo Fisher Scientific Inc. (Waltham, MA, USA). The following antibodies were used for Western blot analysis: polyclonal anti-Bak antibody (#12105), polyclonal anti-Bax antibody (#5023), polyclonal anti-Bim antibody (#2819), monoclonal anti-BNIP3 antibody (#44060); monoclonal anti-BNIP3/NIX antibody (#12396); polyclonal anti-caspase-3 antibody (#9662); polyclonal anti-GPX4 antibody (#52455); polyclonal anti-LC3 antibody (#3868); monoclonal anti-PINK1 antibody (#6946) and monoclonal anti-XBP-1s (#D2C1F) antibody from Cell Signaling Technology, (Leiden, The Netherlands); polyclonal anti-Κ-Casein antibody (#ab111406) from Abcam (Cambridge, UK); polyclonal anti-actin antibody (#A2103) from Sigma-Aldrich (Taufkirchen, Germany ). Monoclonal anti-Bcl-2 antibody (#sc-7382), horseradish peroxidase (HRP)-conjugated goat anti-mouse IgG1, goat anti-rabbit and rabbit anti-goat were purchased from Southern Biotech (Birmingham, AL, USA).

### 2.2. Cell Culture

Human breast adenocarcinoma cells BT549, T47D, and SKBR3 (kindly provided by Prof. Dagmar Kulms, Department of Dermatology, Dresden, Germany) were maintained in RPMI 1640 (Thermo Fisher Scientific Inc., Waltham, MA, USA), supplemented with 10% heat-inactivated fetal calf serum and 1% penicillin-streptomycin in 5% CO_2_. Human breast adenocarcinoma cells MDA-MB-231 (DSMZ, Braunschweig, Germany) were maintained in Leibovitz L15 media (Gibco™, Carlsbad, CA, USA), supplemented with 10% heat-inactivated fetal calf serum, 1% penicillin-streptomycin in 5% CO_2_.

### 2.3. Cell Viability Measurements by ATP and Metabolic (MT) Assays

1.2 × 10^4^ MDA-MB-231, BT549, SKBR3, or T47D cells were seeded into each well of 96-well plates one day prior to treatment. Cells were stimulated in a volume of 50 µL medium. For the cell viability assay, following the manufacturer’s instructions (CellTiter-Glo^®^ Luminescent Cell Viability Assay, Promega, Walldorf, Germany), measurements were performed by adding 50 µL CellTiter-Glo^®^ solution to each well. The luminescence signal was analyzed in duplicates using the microplate reader Infinite M200pro (Tecan, Switzerland). The values were normalized to the viability of untreated cells. One relative unit (RU) corresponds to the viability of untreated cells.

For the metabolic assay, following the manufacturer’s instructions (RealTime-Glo™ MT cell Viability Assay, Promega, Walldorf, Germany), measurements were performed via the addition of 50 µL metabolic substrate (2% MT cell viability substrate and 2% NanoLuc™Enzyme) to each well, directly before stimulation. The luminescence signal was analyzed as above.

### 2.4. Cytotoxicity Measurements by Lactatedehydrogenase (LDH) Assay

1.2 × 10^4^ MDA-MB-231, BT549, SKBR3, or T47D cells were seeded into each well of 96-well plates one day prior to treatment. Cells were stimulated in a volume of 50 µL medium. LDH assay was performed following the manufacturer’s instructions (LDH-Glo^®^ Cytotoxicity Assay, Promega, Walldorf, Germany). In brief, 2 µL cell supernatant was added to 198 µL LDH storage buffer (200 mM Tris HCl, pH 7.3, 10% glycerol, 1% BSA). Then, 50 µL of this solution was mixed with 50 µL LDH-Substrate. The luminescence signal was analyzed as above.

### 2.5. Caspase-3/7 Activity Assay

1.2 × 10^4^ BT549, SKBR3, or T47D cells were seeded into each well of 96-well plates one day prior to treatment. The cells were stimulated in a volume of 50 µL medium. The assay was performed following the manufacturer’s instructions (Caspase-Glo^®^ 3/7 Assay, Promega, Walldorf, Germany). In brief, measurements were performed via the addition of 50 µL of the Caspase-Glo^®^3/7 solution to each well. The luminescence signal was analyzed as above.

### 2.6. Western Blot Analysis

1.25 × 10^6^ BT549, SKBR3, or T47D cells were seeded into each well of 6 well plates one day prior to treatment. Cells were harvested, washed with PBS, lysed for 30 min on ice in lysis buffer (20 mM Tris HCl, pH 7.4, 137 mM NaCl, 2 mM EDTA, 10% glycerine, 1% Triton X-100, Protease Inhibitor mix (Roche, Mannheim, Germany)) and subjected to Western blot analysis. SDS-PAGE was performed with 1% SDS gels. The TransBlot Turbo system (Biorad, Hercules, CA, USA) was used to blot the gels onto nitrocellulose membranes. Blots were blocked with 5% non-fat dried milk in PBS with 0.05% Tween 20 for one hour. Washing steps were performed with PBS-Tween three times for 5 min. Incubation with primary antibodies was performed overnight at 4 °C in PBS-T. HRP-coupled isotype-specific secondary antibodies were incubated for one hour at room temperature in 5% non-fat dried milk. The chemiluminescence signal was produced with LuminataForte (MerckMillipore, Darmstadt, Germany) and detected with a ChemiDoc imaging system (Biorad, Hercules, CA, USA).

### 2.7. IC_50_ Value and CI

GraphPad Prism (version 8.3.0) (GraphPad Software Inc., Boston, MA, USA) software was used to calculate the IC_50_ values for the RL2 and DXR treatments of BT549 cells in the ATP assay over a 24 h period. The value was calculated with concentrations, which are not transformed to logarithms and have a variable slope. The CI (combinatorial index) was calculated using the response additivity [22]. Classification of synergistic, additive, or antagonistic effects was based on the following values: CI > 1 (antagonistic); CI = 1 (additive); CI < 1 (synergistic). Synergisms were calculated according to the Loewe Additivity formula [22]: $\frac{Dosis\;A}{{IC}_{50}\;A}+\frac{Dosis\;B}{{IC}_{50}\;B}=C$.

### 2.8. Statistics

GraphPad Prism (Version 8.3.0) (GraphPad Software, Inc., Boston, MA, USA) software was used to calculate and perform statistic tests. The whole data set of experiments was analyzed using an ordinary one-way ANOVA test. The comparison of specific data sets with each other was performed using the Tukey test. *p*-values were based on the following pattern: ns (not significant; *p* > 0.05), * (significant; *p* < 0.05), ** (significant; *p* < 0.01), *** (significant; *p* < 0.005), **** (significant; *p* < 0.001).

### 2.9. Bioinformatic Analysis

GO (Gene Ontology) term enrichment analysis was carried out using the GONet web server (https://tools.dice-database.org/GOnet/ accessed 18 November 2023) using default settings (Bjoern Peters Lab, La Jolla Institute for Allergy & Immunology 9420 Athena Circle La Jolla, CA 92037, USA) [23]. Gene expression in each cell line was extracted from the CCLE Cell Line Gene Expression Profiles datasets (Expression Public 23Q) [24,25] using the DepMap web server (https://depmap.org/portal/ accessed 18 November 2023) (Broad Institute, Cambridge, MA, USA). The “Harmonizome 3.0” web server was used to extract genes with differential expression in one cell line compared to other cell lines from the CCLE Cell Line Gene Expression Profiles dataset [26].

## 3. Results

### 3.1. RL2 Induces Mitophagy in BT549 and T47D Breast Cancer Cells

The breast cancer cell lines of three different backgrounds were used in the study: BT549 and MDA-MB-231 cells are classified as a model for TNBC, T47D cells are ERα-positive, and SKBR3 cells are HER2-positive. Previously, it has been shown that RL2 treatment induces mitophagy in breast cancer cells, in particular in the MDA-MB-231 cell line (Figure 1a) [21]. Following these findings, it was first tested if RL2 also induces mitophagy in BT549, T47D, and SKBR3 cells. For this purpose, these three cell lines were treated with 300 µg/mL of RL2. The selected concentration of RL2 was chosen based on our previous studies [19,21], in which treatment with this amount of RL2 led to mitophagy and elimination of breast cancer cells. In BT549 cells, treatment with RL2 led to the appearance of key mitophagy markers, as shown by the Western blot. Indeed, the highest levels of the mitophagy marker proteins LC3 II, BNIP3, PINK1, and NIX were observed in this cell line three hours after RL2 treatment (Figure 1b and Appendix A). The appearance of LC3 II upon RL2 treatment was also detected in T47D cells, albeit to a smaller degree in comparison to BT549 cells (Figure 1b and Appendix A). Furthermore, a much smaller increase in BNIP3 levels was observed in T47D cells compared to BT549 cells. In SKBR3 cells, no RL2-induced LC3 conversion was detected (Figure 1b and Appendix A). In SKBR3 cells, almost no increase in PINK1 and BNIP3 levels was observed, as well as changes in NIX levels upon RL2 treatment were not detected. Hence, this indicates that RL2 induced mitophagy in T47D and BT549, with the strongest effects being observed in BT549 cells.

Mitophagy and endoplasmatic reticulum (ER) stress response were shown to be interconnected [27,28]. To determine whether ER stress might play a role in RL2-mediated effects, we examined XBP1-s levels after RL2 administration. XBP1-s is a marker of ER stress. However, no increase in the level of XBP1-s could be observed after RL2 treatment in the cell lines tested (Figure 1b and Appendix A). The addition of RL2 to breast cancer cells was also reported to cause an increase in ROS production, which could lead to the induction of ferroptosis [20,29]. To test whether RL2 treatment induced ferroptosis in the breast cancer cell lines tested, we monitored changes in one of the ferroptosis markers, GPX4. No decrease in GPX4 levels upon RL2 administration could be detected in any of the cell lines, indicating that RL2-mediated signaling is independent of ferroptosis (Figure 1b and Appendix A). RL2 treatment led to mitophagy, especially prominent in BT549 cells, but no induction of ER stress response and ferroptosis was detected in the cell lines tested.

### 3.2. RL2 Treatment Leads to the Viability Loss of BT549 and T47D Breast Cancer Cells

RL2 induced mitophagy and viability loss of MDA-MB-231 cells [19,21]. Next, it was determined whether RL2 induced a loss of viability in BT549, T47D, and SKBR3 cells. The viability of the cells was monitored by two approaches: the measurement of the level of ATP loss as well as the measurement of cellular metabolism. In these assays, we used not only 300 µg/mL but also lower 100 µg/mL and higher 500 µg/mL concentrations of RL2. A decrease in viability was detected in the BT549 and T47D cells after RL2 treatment for 6 and 24 h by measuring ATP loss (Figure 1c,d). Specifically, 100 µg/mL of RL2 had no effect, but the ATP loss was observed starting with treatments of 300 and 500 µg/mL RL2. The metabolic assays also demonstrated a significant reduction in cellular metabolism, but only after 24 h in BT549 and T47D cells (Figure 1f,g). In contrast, nearly no reduction in cell viability upon RL2 treatment was detected in the SKBR3 cells using both assays (Figure 1e,h). Even treatment with high concentrations of RL2 and long stimulation periods did not lead to significant effects on cell viability in this cell line.

To examine whether the differences in sensitivity to RL2 were based on the expression of the pro-apoptotic Bcl-2 family members Bax, Bak, and Bim as well as anti-apoptotic Bcl-2, we analyzed the expression of these proteins using Western blot in BT549, T47D, and SKBR3 cells (Appendix A). This analysis showed the highest expression of Bak and Bim in SKBR3 cells, the most resistant cell line to RL2 treatment. Furthermore, the highest expression of the anti-apoptotic Bcl-2 was observed in the most sensitive cell line to RL2 treatment, BT549. However, it should be noted that Bax was slightly more highly expressed in BT549 cells. This might suggest that activation of Bim and Bak does not play a major role in RL2-mediated cell death and the sensitivity of a particular cell line to RL2 treatment.

Taken together, RL2 treatment induced ATP loss and a decrease in metabolic activity in BT549 and T47D cells, indicating a loss of cell viability upon RL2 treatment. These effects were not observed in SKBR3 cells. Furthermore, RL2 treatment induced mitophagy to a greater extent in BT549 cells. Therefore, we considered BT549 cells as the most RL2-sensitive cell line inducing mitophagy and SKBR3 cells as the RL2-resistant cell line for further analysis.

### 3.3. RL2 Enhances DXR-Induced Cell Viability Loss in BT549, MDA-MB-231 and T47D, but Not in SKBR3 Cells

DXR is a first-line drug in the treatment of breast cancer; however, it has a number of side effects, such as cardiotoxicity. Hence, the development of combination therapies with DXR that allow the reduction in the dose of DXR is of major interest. In this study, we aimed to investigate the RL2 action on DXR-induced cell death in breast cancer cells. First, we measured the ATP loss as the marker of the cell viability reduction upon RL2/DXR treatment for 24 h (Figure 2). Single treatments with 300 µg/mL RL2 or 1.5, 5, and 10 µM DXR reduced the viability of BT549 cells. Notably, RL2/DXR co-treatment resulted in a stronger decrease in cell viability compared to single treatments (Figure 2a,b). Even with lower concentrations of DXR of 1.5 µM, the RL2/DXR co-treatment led to stronger effects compared to single treatments (Figure 2b). To compare these effects to another TNBC cell line, we treated MDA-MB-231 cells with RL2/DXR (Figure 2c). As mentioned, we have previously shown that these cells are very sensitive to RL2 treatment alone and undergo RL2-induced cell death [19,21]. RL2/DXR co-treatment resulted in a strong decrease in viability compared to single treatments in MDA-MB-231 cells (Figure 2c). This demonstrated that RL2 can efficiently sensitize TNBC cells towards DXR treatment.

T47D cells, which are ERα-positive, also showed a stronger decrease in cell viability upon RL2/DXR co-treatment compared to single treatments (Figure 2d). This suggests that not only TNBC cells can be sensitized to DXR treatment by RL2. Strikingly, in SKBR3 cells, DXR administration alone induced a stronger viability loss compared to the BT549, MDA-MB-231, and T47D cells (Figure 2e,f). This was observed already upon treatment with 1.5 µM DXR as well as 5 and 10 µM DXR (Figure 2e,f). However, in contrast to BT549, MDA-MB-231, and T47D cells, RL2 did not enhance the effects of DXR treatment in SKBR3 cells. Hence, in these cells, RL2 treatment had no enhancing effect on the DXR-induced loss of cell viability.

To investigate the effects of RL2/DXR co-treatment on cell viability by the independent approach, we then examined the effects of RL2/DXR co-administration on the metabolic activity of breast carcinoma cells (Appendix A). We selected BT549 cells as the most sensitive and SKBR3 cells as the most resistant for these experiments. Single treatments with RL2 and DXR reduced the metabolic activity of BT549 cells. Combinatorial treatments of RL2 and DXR resulted in an even stronger reduction of metabolic activity in BT549 cells (Appendix A). As expected, RL2 treatment alone did not have any influence on the metabolic activity of SKBR cells. Furthermore, RL2 had no effect on the DXR-dependent reduction of metabolic activity in SKBR3 cells (Appendix A). These results are consistent with the fact that RL2 alone has no effect on SKBR3 cells. RL2 enhanced the DXR-induced loss of viability in BT549, MDA-MB-231, and T47D cells but not in SKBR3 cells.

### 3.4. RL2 Enhances DXR-Induced Caspase-3 Activation and LDH Release

DXR induces intrinsic apoptosis via activation of effector caspases [30]. To test whether RL2 has an effect on DXR-induced caspase-3 activation, we examined caspase-3/7 activity after DXR/RL2 co-treatment as well as DXR or RL2 single treatments in BT549, SKBR3, and T47D cells (Figure 3a–c). In these experiments, we used 5 µM DXR and 300 µg/mL RL2 as conditions leading to the ATP loss and the loss of metabolic activity in BT549 and T47D cells. DXR treatment alone induced caspase-3/7 activity in BT549 and T47D cells as well as in SKBR3 cells. However, it should be noted that the overall level of DXR-induced caspase-3/7 activity was lower in SKBR3 cells than in BT549 and T47D cells. RL2 increased DXR-induced caspase-3/7 activity in the BT549 and T47D cells, consistent with the results on viability loss in these cells. Interestingly, a small increase in caspase-3/7 activity was also observed in the SKBR3 cells upon RL2/DXR addition, but the overall level of caspase-3/7 activity was lower than that in BT549 and T47D cells.

These results were consistent with the Western blot analysis of procaspase-3 processing (Figure 3d and Appendix A). DXR treatment led to the appearance of p19/p17 caspase-3 cleavage products in BT549 cells, the amount of which was slightly increased after RL2/DXR co-treatment. DXR-only treatment led to the appearance of mostly p19-caspase-3 cleavage products in SKBR3 cells, the amount of which was also slightly increased after RL2/DXR co-treatment. Interestingly, the cleavage product of procaspase-3 in SKBR3 cells after both DXR and DXR/RL2 treatment was predominantly p19-caspase-3. Only small amounts of p17-caspase-3 were detected in this cell line (Figure 3d and Appendix A). In contrast, BT549 cells had higher amounts of p17-caspase-3, indicating the generation of the fully catalytically active form of caspase-3 in these cells. It should be noted that the prevalence of catalytically inactive p19-caspase-3 indicates that there is no full autocatalytic processing of caspase-3 in the SKBR3 cells [31,32]. This has been reported to be associated with resistance to apoptotic stimuli and might also explain the results of the caspase-3/7 activity assays, where much less caspase-3/7 activity was detected in the SKBR3 compared to BT549 cells.

These observations were further supported by measurements of LDH release, used as a marker of cell death. This was carried out in BT549, SKBR3, T47D, and MDA-MB-231 cells (Figure 3e–h). In particular, these experiments showed that RL2 enhanced DXR-induced LDH release in BT549, T47D, and MDA-MB-231 cells. In SKBR3 cells, no increase in LDH release was observed with RL2/DXR co-treatment compared to DXR treatment alone. Furthermore, the treatment with RL2 alone did not induce LDH release in SKBR3 cells, further supporting that this cell line is more resistant to the action of RL2 compared to BT549, T47D, and MDA-MB-231 cells (Figure 3f). In conclusion, it was observed that RL2 enhanced the DXR-mediated effects on LDH release in the BT549, T47D, and MDA-MB-231 cells but not in SKBR3 cells.

### 3.5. RL2 and DXR Have Synergistic Effects on Cell Viability Loss

To characterize the effects of RL2 and DXR co-treatment in a quantitative way, we analyzed whether they have synergistic or additive effects. To perform these assays, we used BT549 cells, which were co-treated with different concentrations of RL2 and DXR. Based on these results, the IC_50_ values were calculated for the treatments (Figure 4a,b and Appendix A). The analysis showed that RL2 and DXR have synergistic effects when used at lower concentrations (Figure 4c and Appendix A). The effects of co-treatment with RL2/DXR were then further investigated at an extended concentration range (Figure 4d and Appendix A). Strikingly, this analysis showed that much lower concentrations of DXR can be used to achieve synergistic effects (Figure 4d). It could be shown that even if low concentrations of RL2 and DXR do not affect the viability loss of BT549 cells, combinatorial treatment with these concentrations leads to a significant loss of cell viability, which further supports the synergistic effects of this co-treatment (Figure 4d,e). This analysis demonstrated that RL2 can enhance low-dose DXR-induced loss of cell viability in breast cancer cells, suggesting therapeutic applications for this co-treatment regimen.

## 4. Discussion

The development of antitumor therapies that are specific to a particular type of cancer is urgently needed in modern cancer research. Bioactive peptides derived from human milk have shown promise in treating breast cancer. Lactaptin, a fragment of Κ-Casein, is one of these peptides [16,17]. Previous studies have shown that lactaptin and its recombinant analog, RL2, have potent tumor-eliminating effects in breast, endometrial, lung, and hepatoma cells while not affecting non-malignant cells such as MCF10a cell line [33,34]. Hence, there is accumulating evidence that RL2 specifically affects breast cancer cells and does not have a cell death-inducing effect on non-malignant breast cells [14]. Furthermore, several approaches have been successfully developed to deliver lactaptin into cancer cells, including the use of oncolytic vaccinia viruses and recombinant NK cells [15,35,36]. This highlights the importance of developing further therapeutic approaches based on RL2 [17,19,20,21] as well as targeting the proteins that interact with RL2, such as TOM70 [19]. One of the potential therapeutic directions of RL2 application involves combinatorial treatments, for example, with inducers of extrinsic apoptosis [21]. However, despite the increasing interest in RL2, the effects of RL2 on the action of chemotherapeutic drugs that induce the intrinsic apoptotic pathway have not yet been studied.

Here, we have demonstrated that RL2 can enhance cell death in combinatorial treatments with drugs inducing the intrinsic apoptosis pathway. In particular, we showed that RL2 can enhance the effects of DXR on the cell death of TNBC and ERα positive breast cancer cells. DXR is a first-line drug for the treatment of breast cancer, but it has a number of severe side effects on cancer patients [11,12,13]. Therefore, the development of approaches using DXR in possible lower concentrations in combination with other drugs is of great importance. In this study, we can show that combined treatment with RL2 represents a promising direction due to the synergistic effects of these two drugs in some breast cancer cells. We showed that combinational treatment of RL2/DXR leads to increased caspase-3/7 activity and increased cell viability loss using even very low concentrations of DXR. This allows us to suggest future translational studies involving these two compounds as well as testing the combinations of RL2 with other drugs acting on the intrinsic apoptosis pathway.

Breast cancer cell lines of three different origins were used in this study: TNBC, ERα-positive, and HER2-positive. Strikingly, the TNBC and ERα positive cells were responsive to RL2 and showed an increased loss of cell viability upon combined treatment with DXR. This was not the case for HER2-positive SKBR3 cells. In contrast, HER2-positive SKBR3 cells were resistant to the effects of RL2. RL2 has been shown to be effective in the elimination of TNBC cells, such as the BT549 and MDA-MB-231 cells [17,19,20,21,37]. Importantly, RL2 treatment induces mitophagy in BT549 cells, as shown here, and in MDA-MB-231 cells, as previously reported [21]. This was monitored by the upregulation of key mitophagy markers and LC3 processing. This suggests that mitophagy plays an important role in the elimination of breast cancer cells. Consistent with this hypothesis, T47D cells were also sensitive to RL2 and showed induction of mitophagy, although to a lesser extent compared to TNBC cells. Furthermore, almost no induction of mitophagy by RL2 was detected in SKBR3 cells. This further supports the key role of mitophagy in the elimination of breast cancer cells by RL2. This suggests that RL2 effects may be counteracted by constitutive activation of signaling pathways in HER2-positive cancer cells [38,39]. In particular, HER2-positive cancer cells are characterized by molecular alterations in PI3K/AKT and MAPK pathways and inhibition of autophagy. This allows us to suggest that combined treatment with RL2 and HER2 blockers such as trastuzumab might sensitize HER2-positive cells towards cell death and should be considered in future studies.

To get more insight into the signaling pathways leading to the resistance to RL2 in the SKBR3 cells, we compared the gene expression profiles of the SKBR3, T47D, MDA-MB-231, and BT549 cell lines. This was performed by extracting high or low-expressed genes relative to other cell lines from the CCLE Cell Line Gene Expression Profiles dataset [24] (Figure 5a). The Harmonizome 3.0 web server was used to extract these gene sets [26]. The comparison for gene sets suggests that the SKBR3, T47D, and BT549 cell lines have different expression profiles, and SKBR3 shares more differentially expressed genes with T47D compared to BT549 (Figure 5b). As a next step, we used the GO (Gene Ontology) term enrichment analysis using the GONet tool to reveal biological processes that could be attributed to the differential expression of these genes [23]. In the case of SKBR3, we found enrichment with several processes related to cellular metabolism (Figure 5c). This is consistent with the above-mentioned effects of HER2 overexpression on the activity of the PI3K/AKT and MAPK pathways. Perturbation of cellular metabolic processes may be one of the reasons for the low sensitivity of SKBR3 cells to RL2 treatment. In contrast, only several GO terms with low significance related to cellular metabolism were identified in case of T47D cell line and no enrichment with these processes was found in the BT549 and MDA-MB-231 cells. In addition, we compared the gene expression levels of cell proliferation signature genes using the CCLE Cell Line Gene Expression Profiles dataset. We found that BT549 cells had a higher expression of these genes compared to SKBR3 and T47D cell lines, but for most of them the expression levels were lower in the case of SKBR3 (Figure 5d).

Our results show that RL2 treatment decreases intracellular ATP, which could lead to metabolic reprogramming of the cells. We have also shown that RL2 reduces the metabolic activity of breast cancer cells, which are sensitive to RL2 administration. It is well known that cancer cells have different metabolic properties [40]. However, the influence of these processes on the sensitivity to anticancer treatment is still the focus of research and has yet to be delineated. In this study, three breast cancer cell lines from different backgrounds were used, which, as shown by bioinformatic analysis, can be distinguished by the activation of different metabolic pathways. Strikingly, these cell lines had different sensitivities to RL2 treatment, which could be explained by their different metabolic states. In particular, TNBC cells, which are sensitive to RL2 treatment, have no disruption of metabolic processes. Considering these data, it could be suggested that the different sensitivity to RL2 treatment could also be explained by the metabolic status of certain cancer cells and should be taken into account when considering co-treatment approaches in future therapies.

## 5. Conclusions

In conclusion, in this study, we have analyzed the interplay between RL2 and the chemotherapeutic drugs that induce intrinsic cell death in breast cancer cells, with a particular focus on the co-treatment with DXR (Figure 6). We have shown that RL2 sensitizes TNBC and ERα-positive breast cancer cells to DXR treatment, which involves the induction of mitophagy (Figure 6). This further supports the important role of RL2 as a promising chemotherapeutic agent. Our results allow us to suggest new avenues for translational applications of RL2 in co-treatment regimens with the drugs acting on the intrinsic apoptosis pathway.

## Figures and Tables

**Figure 1 cells-12-02779-f001:**
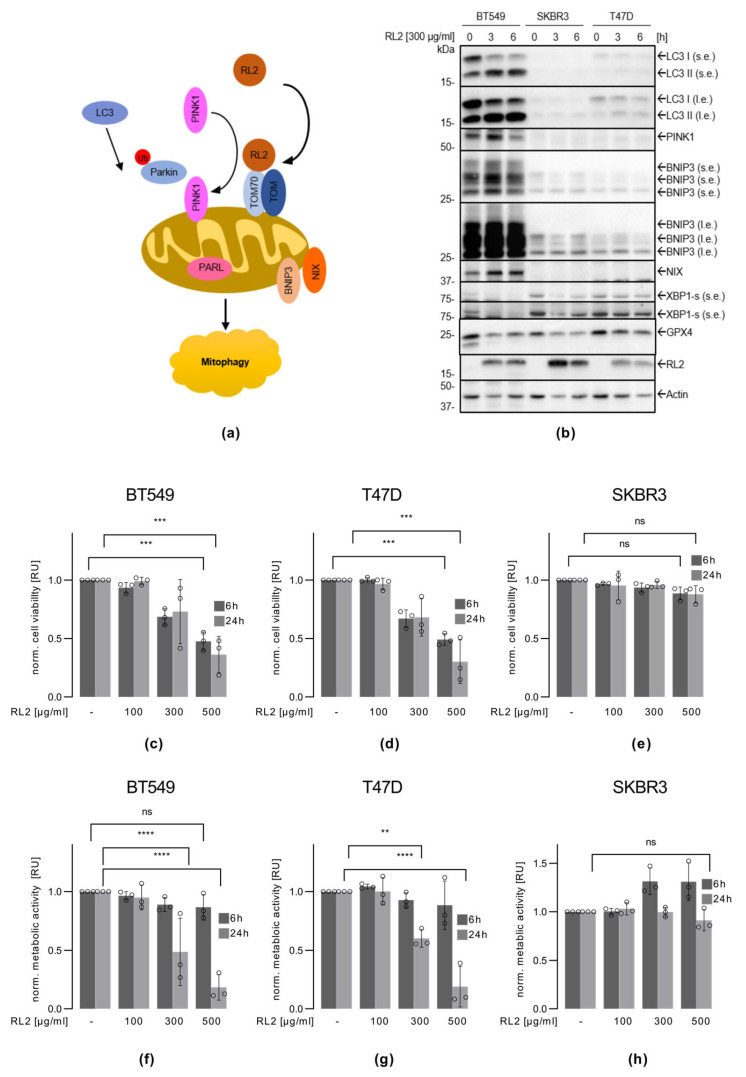
RL2 induces mitophagy and cell viability loss in BT549 and T47D breast cancer cells: (**a**) Scheme of RL2 dependent mitophagy induction. (**b**) BT549, T47D, and SKBR3 cells were treated with 300 µg/mL RL2 for 3 or 6 h or left untreated. Western blot analysis of the indicated proteins. One representative experiment out of three independent ones is shown. (**c**,**f**) BT549, (**d**,**g**) T47D, and (**e**,**h**) SKBR3 cells were treated with RL2 with the indicated concentrations for 6 and 24 h. (**c**–**e**) Cell viability was captured by measuring ATP levels using the CellTiter-Glo^®^ Luminescent Cell Viability Assay. (**f**–**h**) Cell viability was captured by measuring cellular metabolism using RealTime-Glo™ MT cell Viability Assay. (**c**–**h**) Mean and standard deviation are shown for three independent experiments. Statistical analysis was carried out using an ordinary one-way ANOVA with the Tukey test (ns (not significant; *p* > 0.05), ** (significant; *p* < 0.01), *** (significant; *p* < 0.005), **** (significant; *p* < 0.001)).

**Figure 2 cells-12-02779-f002:**
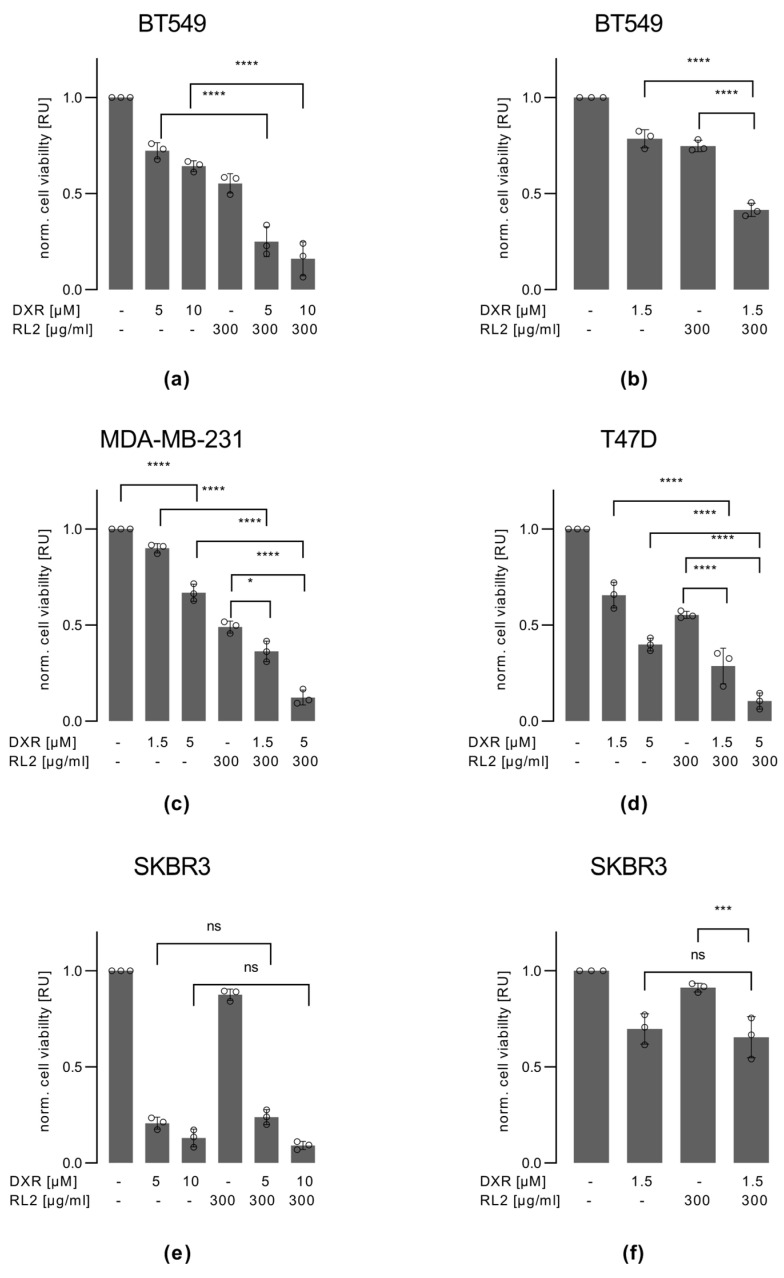
RL2 enhances DXR-induced viability loss in BT549, MDA-MB-231, and T47D but not in SKBR3 cells: (**a**,**b**) BT549, (**c**) MDA-MB-231, (**d**) T47D, and (**e**,**f**) SKBR3 cells were treated with RL2, DXR or both for the indicated concentrations and for 24 h. Cell viability was captured by measuring ATP levels using the CellTiter-Glo^®^ Luminescent Cell Viability Assay. Mean and standard deviation are shown for three independent experiments. Statistical analysis was carried out using an ordinary one-way ANOVA with a Tukey test (ns (not significant; *p* > 0.05), * (significant; *p* < 0.05), *** (significant; *p* < 0.005), **** (significant; *p* < 0.001).

**Figure 3 cells-12-02779-f003:**
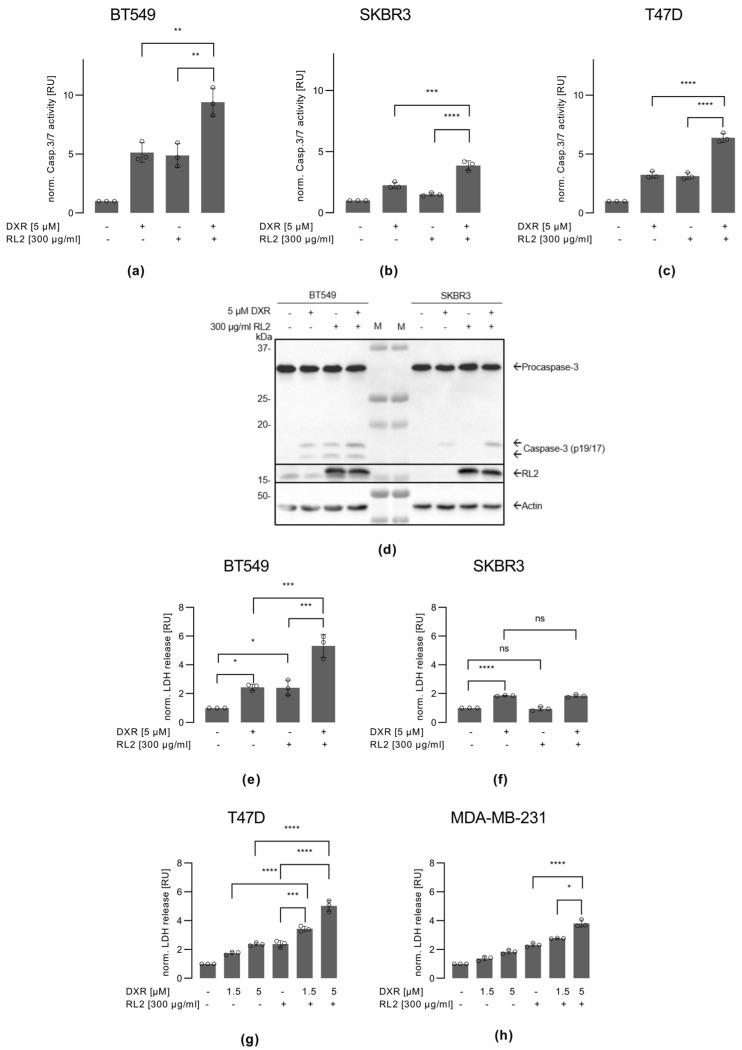
RL2 increases DXR-induced caspase-3 activation and LDH release: (**a**) BT549, (**b**) SKBR3, (**c**) and T47D cells were treated with RL2, DXR, or both for the indicated concentrations for 6 h (**h**). Caspase-3/7 activity was determined using the Caspase-Glo^®^3/7 Assay. (**d**) BT549 and SKBR3 cells were treated with 300 µg/mL RL2, 5 µM DXR, or both for 6 h or left untreated. Western blot analysis of the indicated proteins. One representative experiment out of three independent ones is shown. M: molecular weight protein marker, which was loaded in the middle of the gel two times between the lysates of BT549 and SKBR3 cells. (**e**–**h**) LDH release was measured in (**e**) BT549, (**f**) SKBR3, (**g**) T47D, and (**h**) MDA-MB-231 cells after 24 h treatment with DXR, RL2, or their combination in the indicated concentrations. LDH release is measured using the LDH-Glo^®^ Cytotoxicity Assay. (**a**–**c**; **e**–**h**) Mean and standard deviation are shown for three independent experiments. Statistical analysis was carried out using an ordinary one-way ANOVA with a Tukey test (ns (not significant; *p* > 0.05)), * (significant; *p* < 0.05), ** (significant; *p* < 0.01), *** (significant; *p* < 0.005), **** (significant; *p* < 0.001). Abbreviations: LDH, lactatedehydrogenase; h., hours.

**Figure 4 cells-12-02779-f004:**
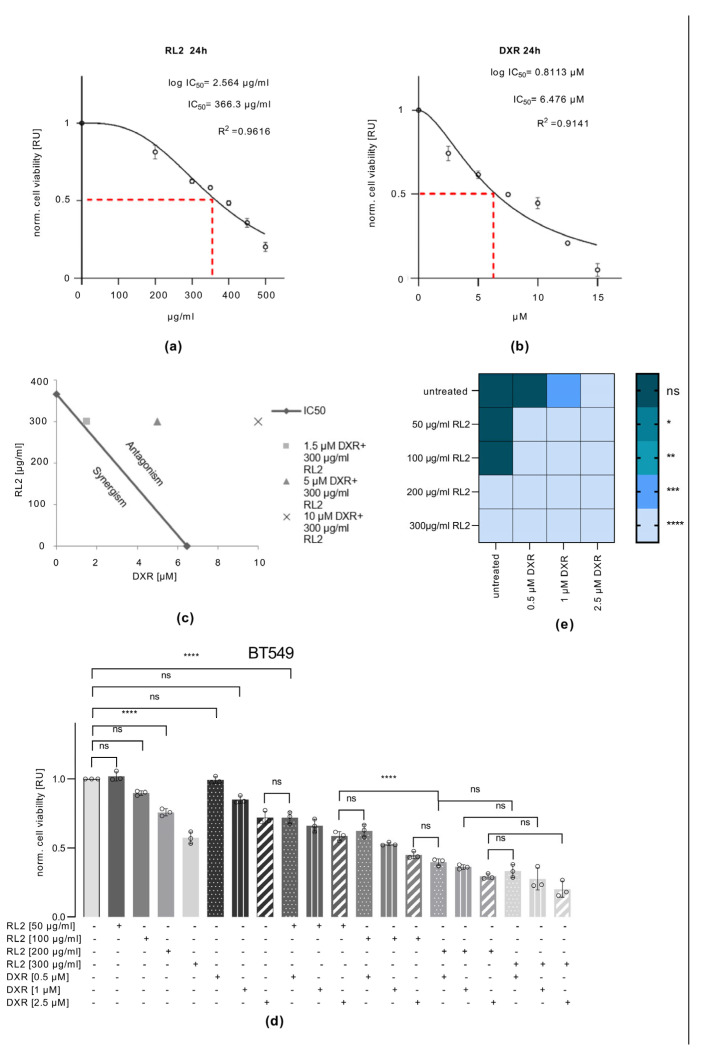
RL2 and DXR have synergistic effects: BT549 cells were treated with RL2, DXR, or both with the indicated concentrations for 24 h. IC_50_ values are calculated for RL2 (**a**) or DXR (**b**) treatment. (**c**) Calculating the area of synergistic, additive, and antagonistic effects using the Loewe additivity index. Shown is the isobologram of the concentrations used in experiments from Figure 2. (**d**) Cells were treated with the indicated concentrations of RL2, DXR, or combinatorial treatment for 24 h. (**a**,**b**,**d**) Cell viability was captured by measuring the ATP levels using the CellTiter-Glo^®^ Luminescent Cell Viability Assay. Mean and standard deviation are shown for three independent experiments. Statistical analysis was carried out using an ordinary one-way ANOVA with a Tukey test: ns (not significant; *p* > 0.05), **** (significant; *p* < 0.001). Bar colours: black edged = untreated; dark grey = 50 µg/mL RL2; middle grey = 100 µg/mL RL2; lighter grey = 200 µg/mL RL2; lightest grey = 300 µg/mL RL2; points = 0.5 µM DXR; vertical lines = 1 µM DXR; diagonal lines = 2.5 µM DXR (**e**) Heat map for the *p*-values calculated from (**d**) *p*-values are compared between each treatment. Dot colors represent the *p*-value shown on the bar: dark green/blue = ns (not significant; *p* > 0.05); dark cyan = * (significant; *p* < 0.05), light cyan ** (significant; *p* < 0.01), blue = *** (significant; *p* < 0.005); light blue = **** (significant; *p* < 0.001).

**Figure 5 cells-12-02779-f005:**
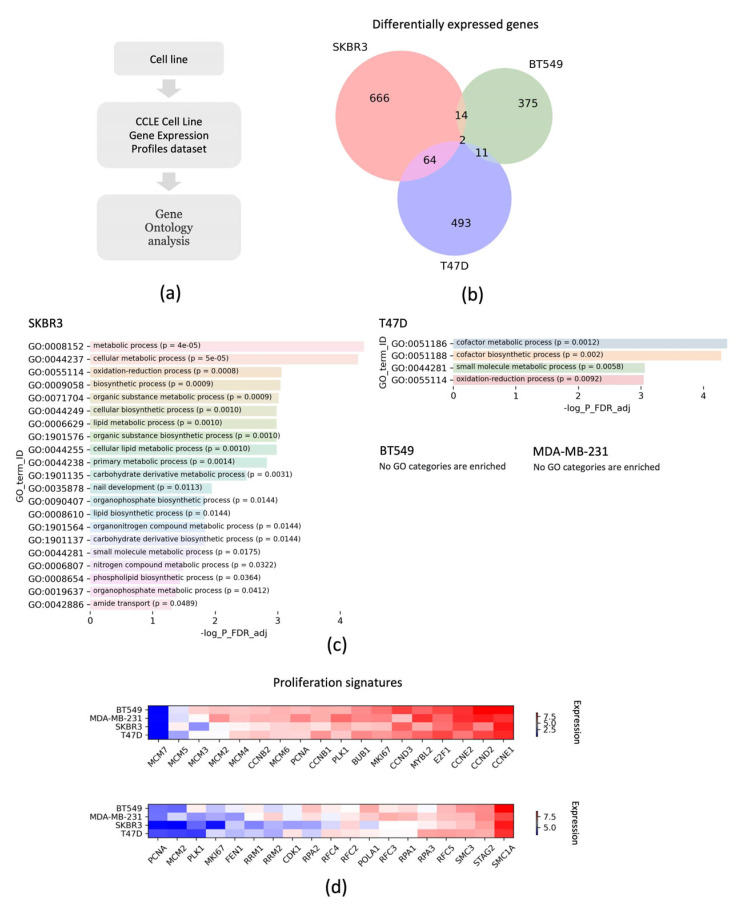
Comparison of gene expression profiles for the SKBR3, T47D, BT549, and MDA-MB-231 cell lines using public databases: (**a**) The scheme of analysis is shown. This includes the selection of the cell lines for comparison with the subsequent mining of the CCLE Cell Line Gene Expression Profiles dataset [24] to extract differentially expressed genes. The “Harmonizome 3.0” web server was used to derive the gene sets. (**b**) Venn diagram of the intersection of differentially expressed genes between the SKBR3, T47D, and BT549 cell lines. (**c**) Gene Ontology (GO) (https://tools.dice-database.org/GOnet/ accessed the 25 November 2023) term enrichment analysis for the differentially expressed genes. GO process names with statistically significant higher expressions are shown for the SKBR3 (top) and T47D (bottom) cell lines; no enrichment was found for the BT549 and MDA-MB-231 cell lines. GO process names and FDR-adjusted *p*-values are indicated. The predictions were performed using the GONet tool [23] (**d**) Comparison of the proliferation signatures of gene expression in SKBR3, T47D, MDA-MB-231, and BT549 cells. Gene expression was derived from the Expression Public 23Q database using the DepMap web server (https://depmap.org/portal/; accessed 18 November 2023) (Broad Institute, Cambridge, MA, USA) ([24]. log_2 (TPM-normalized expression values + 1) are shown in a color gradient from low (blue) to high (red).

**Figure 6 cells-12-02779-f006:**
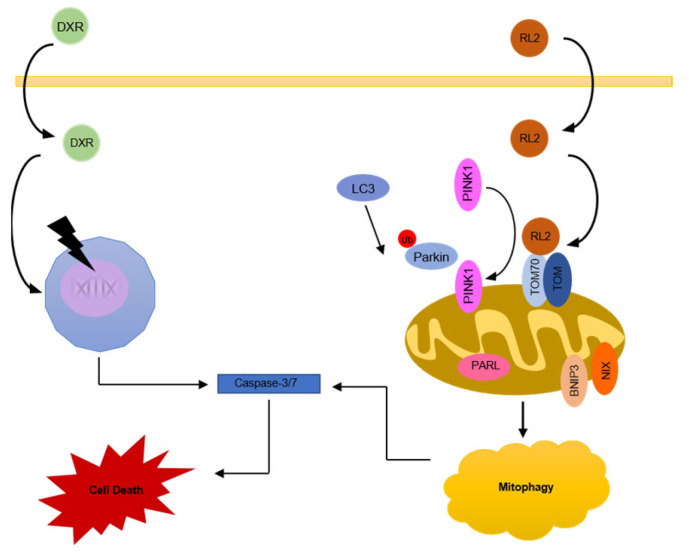
RL2 networks in breast cancer cells: Scheme of RL2-mediated mitophagy and its interplay with DXR-dependent cell death.

## Data Availability

Data are given within the article and Appendix A.

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
