# Peer review of "RL2 Enhances the Elimination of Breast Cancer Cells by Doxorubicin"

_cells, 2023, doi:10.3390/cells12242779_

Round 1

Reviewer 1 Report

Comments and Suggestions for Authors

RL2 enhances the elimination of breast cancer cells by doxorubicin

In present manuscript authors have tested the apoptosis inducing cytotoxic effect of RL2 (recombinant lactaptin 2) on various breast cancer cell lines. The study also experimentally demonstrated the synergistic effect of RL2 with doxorubicin on cell death of BT549 TNBC cell line through apoptosis even at lower dose of doxorubicin. Thus, providing a therapeutic option as combinatorial treatment to avoid doxorubicin mediated side effects in patients. The hypothesis of the current study is very interesting, however, the experiments supporting the claims are not sufficient. Authors need provide some logical clarification and more experimental evidence to be considered for publication.

1.     Please include normal cell control i.e. MCF10a to check the effect of RL2 on normal non-cancer cells as negative control. If possible, please include more than one cell line model to support the claim. Observing the effect in one cell type is not good enough to consolidate the hypothesis that all TNBC will respond the same way. Please replicate the finding in at least one more suitable TNBC cell line model such as HCC1937, LM2 etc.

2.     A lot of similar studies presented in this manuscript regarding the role of RL2 on mitophagy or on cell death of breast cancer cells by RL2 has been previously published in detail by the authors. Despite, the combinatorial studies of RL2 with DXR are interesting. It is recommended to test the effect of this combination therapy using in vivo murine model on tumor growth and to test expression of cleaved caspase 3 in the tumors of treatment vs. control group to strengthen the claim.

3.     Authors mentioned that the use of RL2 in combination with DXR could be a potential therapy in reducing the DXR mediated cytotoxic side effects. Keeping this in mind, the IC50 of the ideal candidate drug compound should be lower than 10 microgram/ml for in vitro studies, however, effective IC50 for RL2 used in this study is ≥300 microgram/ml that is far greater than acceptable. How will authors justify the potential of RL2 to be a future therapeutic option? Please discuss.

4.     Figure S1, S2 and S3 legend says corresponding figures for Fig1h. Should it be Fig1b instead? Please confirm and rectify.

5.     Please discuss plausible explanation of SKBR3 resistance to RL2 in detail.

6.     p-values given in material method section statistics does not match with the p-values given in figure legend. Please confirm and correct.

7.     Line 188 and 189 in manuscript result section are just repeating the already said result.

8.     Poor image quality for all the main figures. Please provide the better resolution images.

9.     Correct the labelling of figure panel D, F, G in Figure 1.

Author Response

Reviewer 1

Major comments  

  1. Please include normal cell control, i.e. MCF10a to check the effect of RL2 on normal non-cancer cells as negative control.If possible, please include more than one cell line model to support the claim. Observing the effect in one cell type is not good enough to consolidate the hypothesis that all TNBC will respond the same way. Please replicate the finding in at least one more suitable TNBC cell line model such as HCC1937, LM2 etc. 

Answer 1:

The experiments on the treatment of non-malignant human cells, e.g. RL2 treatment of non-malignant human mesenchymal stem cells (MSC), were performed in the publication by Semenov et al., Protein Journal 2010. Non-malignant human mesenchymal stem cells (MSC) were completely resistant to the action of RL2. This is one of the first publications on RL2 and all subsequent studies have been based on these results. In order to highlight these findings, as we fully agree that this is very important information, we have included a more prominent citation of this manuscript in the Introduction. Furthermore, the effects of lactaptin on MCF-10A cells were tested in the publication by Kochneva et al. (2018). In this publication, a double recombinant vaccinia virus (VV-GMCSF-Lact) expressing exogenous proteins: the antitumour protein lactaptin and human granulocyte-macrophage colony-stimulating factor (GM-CSF) was constructed and shown to induce loss of cell viability in MDA-MB-231 cells but not in MCF10a cells.  We have also added this citation and reworded the relevant parts of the Introduction and Discussion.

Please, see the new wording below, Introduction, line 55:

…Based on lactaptin different analogs including RL1 (recombinant lactaptin 1) or RL2 (recombinant lactaptin 2) have been generated and shown to induce cell death in breast carcinoma cells but not in the non-cancerous cells [14, 15] with RL2, which consists of amino acids 23-134 of human Κ–casein, being the strongest inducer of cell death [14]. RL2 treatment can lead to cell death in different breast carcinoma cells but not in normal cells and inhibit tumor growth in mouse models [16, 17].

Discussion, line 532:

…Previous studies have shown that lactaptin and its recombinant analogue RL2, have potent tumor-eliminating effects in breast, endometrial, lung and hepatoma cells, while not affecting non-malignant cells such as MCF10a cell line [33, 34]. Hence, there is accumulating evidence that RL2 specifically affects breast cancer cells and does not have a cell death inducing effect on non-malignant breast cells [14].

For the use of another TNBC cell line: To follow the reviewer's suggestion, we have added here the experiments with MDA-MB-231 cells, which are also TNBC. These cells were also sensitised to doxorubicin treatment by RL2. Please see the new Figures 2c and 3h. The detailed characterisation of the effects of RL2 on MDA-MB-231 cells has been previously reported by us (Richter et al., 2020; Wohlfromm et al., 2021). Therefore, this study further extends our knowledge of the effects of RL2 in MDA-MB-231 cells and shows that they can be co-sensitised to doxorubicin treatment by RL2. We have also rewritten several parts of the manuscript to emphasise the use of this cell line in many instances. We thank the reviewer for this important comment.

  1. A lot of similar studies presented in this manuscript regarding the role of RL2 on mitophagy or on cell death of breast cancer cells by RL2 has been previously published in detail by the authors. Despite, the combinatorial studies of RL2 with DXR are interesting. It is recommended to test the effect of this combination therapy using in vivomurine model on tumor growth and to test expression of cleaved caspase 3 in the tumors of treatment vs. control group to strengthen the claim.

Answer 2:

 We fully agree that in vivo experiments are very important. However, we also think that this should be the subject of a follow-up study that we are currently planning.

At the same time, in order to demonstrate the potential of this direction for translational studies, we believe it is very important to publish the effects of RL2 in co-stimulation with the intrinsic pathway inducers on cell lines in its current form. We supported this point of view by adding additional phrasing on this.

Please, see the discussion, line 539:

…This highlights the importance of developing further therapeutic approaches based on RL2 [17, 19-21] as well as targeting the proteins that interact with RL2 such as TOM70 [19]. One of the potential therapeutic directions of RL2 application involves combinatorial treatments, for example with inducers of extrinsic apoptosis [21]. However, despite the increasing interest in RL2, the effects of RL2 on the action of chemotherapeutic drugs that induce the intrinsic apoptotic pathway have not yet been studied.

Here, we have demonstrated that RL2 can enhance cell death in combinatorial treatments with drugs inducing the intrinsic apoptosis pathway. In particular, we showed that RL2 can enhance the effects of DXR on the cell death of TNBC and ERα positive breast cancer cells…

  1. Authors mentioned that the use of RL2 in combination with DXR could be a potential therapy in reducing the DXR mediated cytotoxic side effects. Keeping this in mind, the IC50 of the ideal candidate drug compound should be lower than 10 microgram/ml for in vitrostudies, however, effective IC50 for RL2 used in this study is ≥300 microgram/ml that is far greater than acceptable. How will authors justify the potential of RL2 to be a future therapeutic option? Please discuss.

Answer 3:

We fully support this comment by the reviewer. To overcome this obstacle, as well as the problems associated with the degradation of recombinant RL2 when added to cells (see Figure 1 in Richter et al, 2020), several other therapeutic approaches of RL2 delivary into the cells are under development. The first is the use of oncolytic vaccinia viruses to deliver lactaptin into cells, as mentioned above. In addition, the second approach is the construction of recombinant NK cells. We have added the relevant text and citations to the Discussion section of the manuscript.

Please see the text below, line 536:

…Furthermore, several approaches have been successfully developed to deliver lactaptin into cancer cells, including the use of oncolytic vaccinia viruses and recombinant NK cells [15, 35, 36]…

  1. Figure S1, S2 and S3 legend says corresponding figures for Fig1h. Should it be Fig1b instead? Please confirm and rectify.

Answer 4:

This issue was fixed. We thank the reviewer for this comment and apologize for this typo.

  1. Please discuss plausible explanation of SKBR3 resistance to RL2 in detail.

Answer 5:

We expanded the discussion of SKBR3 resistance. Please, see the new text below,

Discussion, line 558:

..Breast cancer cell lines of three different origins were used in this study: TNBC, ERα positive and HER2 positive. Strikingly, the TNBC and ERα positive cells were responsive to RL2 and showed an increased loss of cell viability upon combined treatment with DXR. This was not the case for HER2-positive SKBR3 cells. In contrast, HER2-positive SKBR3 cells were resistant to the effects of RL2. RL2 has been shown to be effective in the elimination of TNBC cells such as the BT549 and MDA-MB-231 cells [17, 19-21, 37]. Importantly, RL2 treatment induces mitophagy in BT549 cells as shown here and in MDA-MB-231 cells as previously reported [21]. This was monitored by upregulation of key mitophagy markers and LC3 processing. This suggests that mitophagy plays an important role in the elimination of breast cancer cells. Consistent with this hypothesis, T47D cells were also sensitive to RL2 and showed induction of mitophagy, although to a lesser extent compared to TNBC cells. Furthermore, almost no induction of mitophagy by RL2 was detected in SKBR3 cells. This further supports the key role of mitophagy in the elimination of breast cancer cells by RL2. This suggests that RL2 effects may be counteracted by constitutive activation of signaling pathways in HER2-positive cancer cells [38, 39]. In particular, HER2-positive cancer cells are characterized by molecular alterations in PI3K/AKT and MAPK pathways and inhibition of autophagy. This, allows to suggest that combined treatment with RL2 and HER2 blockers such as trastuzumab might allow to sensitize HER2 positive cells towards cell death and should be considered in the future studies…

  1. p-values given in material method section statistics does not match with the p-values given in figure legend. Please confirm and correct.

Answer 6:

This issue has been fixed. We thank the reviewer for this comment and apologise for the previous inconsistencies. 

  1. Line 188 and 189 in manuscript result section are just repeating the already said result.

Answer 7:

This paragraph has been largely rewritten. We now discuss the effects on each cell line sequentially: BT549, T47D and then SKBR3 cells.

Please, see the new text, line 192:

…In BT549 cells, treatment with RL2 led to the appearance of key mitophagy markers as shown by Western Blot. Indeed, the highest levels of the mitophagy marker proteins LC3 II, BNIP3, PINK1 and NIX were observed in this cell line three hours after RL2 treatment (Figure 1b, Figures S1, S2). The appearance of LC3 II upon RL2 treatment was also detected in T47D cells albeit to the smaller degree in comparison to BT549 cells (Figure 1b, Figures S1, S2). Furthermore, a much smaller increase in BNIP3 levels was observed in T47D cells compared to BT549 cells. In SKBR3 cells no RL2-induced LC3 conversion was detected (Figure 1b, Figures S1, S2). Furthermore, in SKBR3 cells, almost no increase in PINK1 and BNIP3 levels was observed, as well as changes in NIX levels upon RL2 treatment were not detected. Hence, this indicates that RL2 induced mitophagy in T47D and BT549, with the strongest effects being observed in BT549 cells…

  1. Poor image quality for all the main figures. Please provide the better resolution images.

  1. Correct the labelling of figure panel D, F, G in Figure 1.

Answer 8-9:

All the issues were fixed. We thank the reviewer for these comments.

General answer:

We would like to thank the reviewer for all the excellent comments and for taking the time to carefully review our manuscript and provide a number of very valuable comments.

Reviewer 2 Report

Comments and Suggestions for Authors

The authors have described the role of RL2 in enhancing the pharmacological role of Doxorubicin in Breast cancer cell lines, namely, BT549, T47D, and SKBR3. The study also highlights the role of RL2 in enhancing the intrinsic apoptotic pathway. The manuscript is well-written and scientifically sound. However, if possible, the authors need to address and emphasize the role of mitochondria in the RL2 - Doxorubicin treatment.

1. Do Bax and Bim have a role in inducing apoptosis?

2. If yes, does their expression vary in the three cell lines?

3. Any disruption in Oxygen consumption or glycolytic pathway upon treatment with RL2 and DXR?

Author Response

Rev 2

 The authors have described the role of RL2 in enhancing the pharmacological role of Doxorubicin in Breast cancer cell lines, namely, BT549, T47D, and SKBR3. The study also highlights the role of RL2 in enhancing the intrinsic apoptotic pathway. The manuscript is well-written and scientifically sound. However, if possible, the authors need to address and emphasize the role of mitochondria in the RL2 - Doxorubicin treatment.

Answer:

We thank the reviewer for the positive feedback on our study.

  1. Do Bax and Bim have a role in inducing apoptosis?
  2. If yes, does their expression vary in the three cell lines? 

Answer 1-2:

We analysed the expression of Bax and BIM in these three cell lines. See new Figures S5 and S6. The results show that BIM is more highly expressed in SKBR3 cells (as is Bak), which largely rules out its role in conferring resistance to RL2 action in this cell line. Importantly, however, these results are consistent with the higher sensitivity of SKBR3 cells to doxorubicin treatment compared to BT549 cells that we observed in our study. Indeed, this cell line loses viability to a greater extent than BT549 cells when treated with the same dose of doxorubicin. This may be consistent with the expression levels of Bim and Bak in these two cell lines.

We have also added the relevant text to the manuscript, line 232:

…To examine whether the differences in sensitivity to RL2 were linked to the expression of the pro-apoptotic Bcl-2 family members Bax, Bak and Bim as well as the anti-apoptotic Bcl-2, we analysed the expression of these proteins by Western Blot in BT549, T47D and SKBR3 cells (Figure S4, S5). This analysis showed the highest expression of Bak and Bim in SKBR3 cells, the most resistant cell line to RL2 treatment. Furthermore, the highest expression of the anti-apoptotic Bcl-2 was observed in the most sensitive cell line to RL2 treatment, BT549. However, it should be noted that Bax was slightly more highly expressed in BT549 cells. This might suggest that activation of Bim and Bak does not play a major role in RL2-mediated cell death and the sensitivity of a particular cell line to RL2 treatment…

Anyhow, we fully agree with the reviewer here and we think that the role of mitochondria should be very multifactorial in this context and future studies ‚beyond Bim and Bax‘ are definitely required.

  1. Any disruption in Oxygen consumption or glycolytic pathway upon treatment with RL2 and DXR?

Answer 3:

We have previously shown that RL2 has no effect on oxygen consumption in Wohlfromm et al, 2020. This allows us to conclude that the effect of RL2 does not directly involve perturbation of the mitochondrial respiratory chain, and other mechanisms leading to the loss of cell viability upon RL2 treatment have been further investigated. Based on this information, we did not address this issue in detail in this manuscript, and following the reviewer's thoughtful comment, we highlight this issue in the Discussion. However, we fully agree with the reviewer that verification of this pathway in the context of potential co-treatment with Doxorubicin is important.

Further, we highlighted our previous findings in the introduction to be clear why we did not take this direction of research in this manuscript, line 68:

…It was also shown that RL2 had no effect on oxygen consumption indicating that the effect of RL2 does not directly involve perturbation of the mitochondrial respiratory chain [21]…

General answer:

We would like to thank the reviewer for important comments and for taking the time to carefully review our manuscript and provide a number of very valuable comments.

Reviewer 3 Report

Comments and Suggestions for Authors

Thank you for inviting me to evaluate the article titled “RL2 enhances the elimination of breast cancer cells by doxorubicin”. This study analysed the effects of RL2 on the doxorubicin (DXR)-induced cell death in breast cancer cells and further explored the molecular mechanisms of RL2 action on intrinsic pathway. The contribution of these observation to the related field is novel and the authors describe their results rationale. I have some minor comments as below.

1.      In 3.1., the author mentioned that “T47D and SKBR3 cells, the increase in PINK1 and BNIP3 levels was observed to a lesser extent, while changes in NIX levels upon RL2 treatment were not detected” (Line 190-192). However, I did not observe an increased expression after RL2 treatment of PINK1 and BNIP3 in both cell lines.

2.      The title of this project more focused on all breast cancer cells, but the results showed that only BT549 works best among all of these three cell-lines. Therefore, I suggest the author only mention the TNBC subtypes in the title.

3.      In the discussion part, the author discussed the cell line different among BT549, SKBR3 and T47D (Line 383-393). I suggest they could add more discussion of why the TNBC cell line works better under RL2 treatment compare with non-TNBC cancer subtypes.

4.      The authors also mentioned that “The comparison for gene sets suggests that SKBR3, T47D and BT549 cell lines have different expression profiles”. The authors need to mention where these different expression profiles come from? They compare the differentially expressed genes between cell-lines or what?

5.      The limitations of this research also need to be discussed in the discussion section.

6.      There were some small grammar errors need to be revised, please pay attention to.

Comments on the Quality of English Language

Minor editing of English language required

Author Response

Thank you for inviting me to evaluate the article titled “RL2 enhances the elimination of breast cancer cells by doxorubicin”. This study analysed the effects of RL2 on the doxorubicin (DXR)-induced cell death in breast cancer cells and further explored the molecular mechanisms of RL2 action on intrinsic pathway. The contribution of these observation to the related field is novel and the authors describe their results rationale. I have some minor comments as below.

Answer:

We thank the reviewer for the positive feedback on our study.

  1. In 3.1., the author mentioned that “T47D and SKBR3 cells, the increase in PINK1 and BNIP3 levels was observed to a lesser extent, while changes in NIX levels upon RL2 treatment were not detected” (Line 190-192). However, I did not observe an increased expression after RL2 treatment of PINK1 and BNIP3 in both cell lines.

Answer 1:

We have rephrased this part of the manuscript. In particular, the increase in BNIIP3 in T47D cells was very low, while no increase in PINK1 and NIX could be detected in this cell line. No induction of mitophagy could be detected in SKBR3 cells. We fully agree with the reviewer and thank for this important comment.

Please, see the new text, line 192:

…In BT549 cells, treatment with RL2 led to the appearance of key mitophagy markers as shown by Western Blot. Indeed, the highest levels of the mitophagy marker proteins LC3 II, BNIP3, PINK1 and NIX were observed in this cell line three hours after RL2 treatment (Figure 1b, Figures S1, S2). The appearance of LC3 II upon RL2 treatment was also detected in T47D cells albeit to the smaller degree in comparison to BT549 cells (Figure 1b, Figures S1, S2). Furthermore, a much smaller increase in BNIP3 levels was observed in T47D cells compared to BT549 cells. In SKBR3 cells no RL2-induced LC3 conversion was detected (Figure 1b, Figures S1, S2). Furthermore, in SKBR3 cells, almost no increase in PINK1 and BNIP3 levels was observed, as well as changes in NIX levels upon RL2 treatment were not detected. Hence, this indicates that RL2 induced mitophagy in T47D and BT549, with the strongest effects being observed in BT549 cells…

  1. The title of this project more focused on all breast cancer cells, but the results showed that only BT549 works best among all of these three cell-lines. Therefore, I suggest the author only mention the TNBC subtypes in the title.

Answer 2:

During the revision process, we added another TNBC cell line, MDA-MB-231, which behaved very similarly to BT549 cells. However, we also added T47D cells, which also showed sensitisation. T47D are ERa positive. Therefore, these results preclude the inclusion of TNBC in the manuscript title. However, we would like to emphasise that both TNBC and ERa work, please see our new abstract and conclusions

Abstract, line 13,

RL2 (recombinant lactaptin 2), a recombinant analogon of the human milk protein Κ-Casein induces mitophagy and cell death in breast carcinoma cells. Furthermore, RL2 was shown to enhance extrinsic apoptosis upon long-term treatment while inhibiting it upon short-term stimulation. However, the effects of RL2 on the action of chemotherapeutic drugs that induce the intrinsic apoptotic pathway have not been investigated to date. Here, we examined the effects of RL2 on the doxorubicin (DXR)-induced cell death in breast cancer cells with three different backgrounds. In particular, we used BT549 and MDA-MB-231 triple negative breast cancer (TNBC) cells, T47D estrogen receptor alpha (ERα) positive cells and SKBR3 human epidermal growth factor receptor 2 (HER2) positive cells. BT549, MDA-MB-231 and T47D cells showed a severe loss of cell viability upon RL2 treatment, accompanied by the induction of mitophagy. Furthermore, BT549, MDA-MB-231 and T47D cells could be sensitised towards DXR treatment with RL2, as evidenced by loss of cell viability. In contrast, SKBR3 cells showed almost no RL2-induced loss of cell viability when treated with RL2 alone and RL2 did not sensitise SKBR3 cells towards DXR-mediated loss of cell viability. Bioinformatic analysis of gene expression showed an enrichment of genes controlling metabolism in SKBR3 cells compared to the other cell lines. This suggests that the metabolic status of the cells is important for their sensitivity to RL2. Taken together, we have shown that RL2 can enhance the intrinsic apoptotic pathway in TNBC and ERα-positive breast cancer cells, paving the way for the development of novel therapeutic strategies

Conclusions, line 615:

…In conclusion, in this study we have analysed the interplay between RL2 and the chemotherapeutic drugs that induce intrinsic cell death in breast cancer cells, with the particular focus on the co-treatment with DXR (Figure 6). We have shown that RL2 sensitizes TNBC and ERα positive breast cancer cells to DXR treatment, which involves the induction of mitophagy (Figure 6). This further supports the important role of RL2 as a promising chemotherapeutic agent. Our results allow to suggest new avenues for translational applications of RL2 in co-treatment regimens with the drugs acting on the intrinsic apoptosis pathway.

  1. In the discussion part, the author discussed the cell line different among BT549, SKBR3 and T47D (Line 383-393). I suggest they could add more discussion of why the TNBC cell line works better under RL2 treatment compare with non-TNBC cancer subtypes.

Answer 3:

The bioinformatics analysis revealed some features that can be used to hypothesise the sensitivity of TNBC cell lines to RL2, which we have put into discussion:

Line 605:

…In this study, three breast cancer cell lines from different backgrounds were used, which, as shown by bioinformatic analysis, can be distinguished by the activation of different metabolic pathways. Strikingly, these cell lines had different sensitivities to RL2 treatment, which could be explained by their different metabolic states. In particular, TNBC cells, which are sensitive to RL2 treatment, have no disruption of metabolic processes…

Furthermore, as mentioned in response 2, in the current version of the manuscript we also highlight ERa-positive cells as promising in this setup. Therefore, we also compare all three cell types in the discussion.

Discussion, line 558:

..Breast cancer cell lines of three different origins were used in this study: TNBC, ERα positive and HER2 positive. Strikingly, the TNBC and ERα positive cells were responsive to RL2 and showed an increased loss of cell viability upon combined treatment with DXR. This was not the case for HER2-positive SKBR3 cells. In contrast, HER2-positive SKBR3 cells were resistant to the effects of RL2. RL2 has been shown to be effective in the elimination of TNBC cells such as the BT549 and MDA-MB-231 cells [17, 19-21, 37]. Importantly, RL2 treatment induces mitophagy in BT549 cells as shown here and in MDA-MB-231 cells as previously reported [21]. This was monitored by upregulation of key mitophagy markers and LC3 processing. This suggests that mitophagy plays an important role in the elimination of breast cancer cells. Consistent with this hypothesis, T47D cells were also sensitive to RL2 and showed induction of mitophagy, although to a lesser extent compared to TNBC cells. Furthermore, almost no induction of mitophagy by RL2 was detected in SKBR3 cells. This further supports the key role of mitophagy in the elimination of breast cancer cells by RL2. This suggests that RL2 effects may be counteracted by constitutive activation of signaling pathways in HER2-positive cancer cells [38, 39]. In particular, HER2-positive cancer cells are characterized by molecular alterations in PI3K/AKT and MAPK pathways and inhibition of autophagy. This, allows to suggest that combined treatment with RL2 and HER2 blockers such as trastuzumab might allow to sensitize HER2 positive cells towards cell death and should be considered in the future studies…

  1. The authors also mentioned that “The comparison for gene sets suggests that SKBR3, T47D and BT549 cell lines have different expression profiles”. The authors need to mention where these different expression profiles come from? They compare the differentially expressed genes between cell-lines or what?

Answer 4:

We have largely rewritten and expanded this section. We hope that it is now clearer. In short, the different expression profiles of the cell lines used in the study were taken from the databases and not generated by us. To make this point clearer, we have added the analysis scheme, i.e. the workflow, which clearly shows the source of our data.

Please see the new Figure 5

  1. The limitations of this research also need to be discussed in the discussion section

Answer 5:

We have added the discussion of the potential therapeutic applications of the proposed novel delivery approach using oncolytic vaccinia viruses to deliver lactaptin into cells.

We have added the relevant text and citations to the Discussion section of the manuscript.

Please see the text below, line 536:

…Furthermore, several approaches have been successfully developed to deliver lactaptin into cancer cells, including the use of oncolytic vaccinia viruses and recombinant NK cells [15, 35, 36]…

  1. There were some small grammar errors need to be revised, please pay attention to.

Answer 6:

We have carefully checked the spelling and hope that they are not present in the current version.

General answer:

We would like to thank the reviewer for highly valuable comments and for taking the time to carefully review our manuscript.

Reviewer 4 Report

Comments and Suggestions for Authors

It is not clear how the gene expression study was conducted. Ideally, this should be performed before and after the treatment using RNA-Seq. It appears that Figure 5 was not based on the author’s own analysis.

The quality of the figure is too low.

Why T-47 cells were not included in Figure 2 and Figure 3?

The CellTiter Glo assay results were shown in percentage viability compared to control while the LDH assay were reported in raw LDH release, why? This should be normalized to control and the results should be compared with CellTiter Glo assay.

Comments on the Quality of English Language

Some sentences are too long and hard to read. 

Author Response

  1. It is not clear how the gene expression study was conducted. Ideally, this should be performed before and after the treatment using RNA-Seq. It appears that Figure 5 was not based on the author’s own analysis.
  2. The quality of the figure is too low.

Answer 1-2:

We have largely rewritten and expanded this section. We hope that it is now clearer. In short, the different expression profiles of the cell lines used in the study were taken from the databases and not generated by us. To make this point clearer, we have added the analysis scheme, i.e. the workflow, which clearly shows the source of our data.

Please see the new Figure 5

  1. Why T-47 cells were not included in Figure 2 and Figure 3? 

 Answer 3:

Our strategy was to consider one sensitive and one resistant cell line, but following the reviewer's comment, we have added some results on T47D cells to the manuscript. Please see the new panels: 2d, 3c and 3g, e.g. cell viability assay, caspase-3-7 activity assay and LDH assay, respectively.  These cells were also sensitised to doxorubicin treatment by RL2 as can be observed in these experiments. We have subsequently added this information to many parts of the manuscript, including the Abstract and Conclusions:

Abstract, line 13,

RL2 (recombinant lactaptin 2), a recombinant analogon of the human milk protein Κ-Casein induces mitophagy and cell death in breast carcinoma cells. Furthermore, RL2 was shown to enhance extrinsic apoptosis upon long-term treatment while inhibiting it upon short-term stimulation. However, the effects of RL2 on the action of chemotherapeutic drugs that induce the intrinsic apoptotic pathway have not been investigated to date. Here, we examined the effects of RL2 on the doxorubicin (DXR)-induced cell death in breast cancer cells with three different backgrounds. In particular, we used BT549 and MDA-MB-231 triple negative breast cancer (TNBC) cells, T47D estrogen receptor alpha (ERα) positive cells and SKBR3 human epidermal growth factor receptor 2 (HER2) positive cells. BT549, MDA-MB-231 and T47D cells showed a severe loss of cell viability upon RL2 treatment, accompanied by the induction of mitophagy. Furthermore, BT549, MDA-MB-231 and T47D cells could be sensitised towards DXR treatment with RL2, as evidenced by loss of cell viability. In contrast, SKBR3 cells showed almost no RL2-induced loss of cell viability when treated with RL2 alone and RL2 did not sensitise SKBR3 cells towards DXR-mediated loss of cell viability. Bioinformatic analysis of gene expression showed an enrichment of genes controlling metabolism in SKBR3 cells compared to the other cell lines. This suggests that the metabolic status of the cells is important for their sensitivity to RL2. Taken together, we have shown that RL2 can enhance the intrinsic apoptotic pathway in TNBC and ERα-positive breast cancer cells, paving the way for the development of novel therapeutic strategies

Conclusions, line 615:

…In conclusion, in this study we have analysed the interplay between RL2 and the chemotherapeutic drugs that induce intrinsic cell death in breast cancer cells, with the particular focus on the co-treatment with DXR (Figure 6). We have shown that RL2 sensitizes TNBC and ERα positive breast cancer cells to DXR treatment, which involves the induction of mitophagy (Figure 6). This further supports the important role of RL2 as a promising chemotherapeutic agent. Our results allow to suggest new avenues for translational applications of RL2 in co-treatment regimens with the drugs acting on the intrinsic apoptosis pathway.

  1. The CellTiter Glo assay results were shown in percentage viability compared to control while the LDH assay were reported in raw LDH release, why? This should be normalized to control and the results should be compared with CellTiter Glo assay.

Answer 4:

We have performed these recalculations. Please see the new figures.

General answer:

We would like to thank the reviewer for very valuable comments and for taking the time to carefully review our manuscript.

Round 2

Reviewer 1 Report

Comments and Suggestions for Authors

In the revised manuscript, authors have answered all the queries and updated the results and conclusions of the study accordingly. Data are well-presented and of interest. I have no further revisions.

Reviewer 3 Report

Comments and Suggestions for Authors

The authors addressed most of my concerns; now the paper is more readable.